# BRAIN-LIKE APPROACHES TO UNSUPERVISED LEARNING OF HIDDEN REPRESENTATIONS - A COMPARATIVE STUDY

## ABSTRACT

Unsupervised learning of hidden representations has been one of the most vibrant research directions in machine learning in recent years. In this work we study the brain-like Bayesian Confidence Propagating Neural Network (BCPNN) model, recently extended to extract sparse distributed high-dimensional representations. The saliency and separability of the hidden representations when trained on MNIST dataset is studied using an external linear classifier and compared with other unsupervised learning methods that include restricted Boltzmann machines and autoencoders.

## 1 INTRODUCTION

Artificial neural networks have made remarkable progress in supervised pattern recognition in recent years. In particular, deep neural networks have dominated the field largely due to their capability to discover hierarchies of salient data representations. However, most recent deep learning methods rely extensively on supervised learning from labelled samples for extracting and tuning data representations. Given the abundance of unlabeled data there is an urgent demand for unsupervised or semi-supervised approaches to learning of hidden representations (Bengio et al., 2013). Although early concepts of greedy layer-wise pretraining allow for exploiting unlabeled data, ultimately the application of deep pre-trained networks to pattern recognition problems rests on label dependent end-to-end weight fine tuning (Erhan et al., 2009). At the same time, we observe a surge of interest in more brain plausible networks for unsupervised and semi-supervised learning problems that build on some fundamental principles of neural information processing in the brain (Pehlevan & Chklovskii, 2019; Illing et al., 2019). Most importantly, these brain-like computing approaches rely on local learning rules and label independent biologically compatible mechanisms to build data representations whereas deep learning methods predominantly make use of error back-propagation (backprop) for learning the weights. Although efficient, backprop has several issues that make it an unlikely candidate model for synaptic plasticity in the brain. The most apparent issue is that the synaptic connection strength between two biological neurons is expected to comply with Hebb's postulate, i.e. to depend only on the available local information provided by the activities of pre- and postsynaptic neurons. This is violated in backprop since synaptic weight updates need gradient signals to be communicated from distant output layers. Please refer to (Whittington & Bogacz, 2019; Lillicrap et al., 2020) for a detailed review of possible biologically plausible implementations of and alternatives to backprop.

In this work we utilize the MNIST dataset to compare two classical learning systems, the autoencoder (AE) and the restricted Boltzmann machine (RBM), with two brain-like approaches to unsupervised learning of hidden representations, i.e. the recently proposed model by Krotov and Hopfield (referred to as the KH model) (Krotov & Hopfield, 2019), and the BCPNN model (Ravichandran et al., 2020), which both rely on biologically plausible learning strategies. In particular, we qualitatively examine the extracted hidden representations and quantify their label dependent separability using a simple linear classifier on top of all the networks under investigation. This classification step is not part of the learning strategy, and we use it merely to evaluate the resulting representations.

Special emphasis is on the feedforward BCPNN model with a single hidden layer, which frames the update and learning steps of the neural network as probabilistic computations. Probabilistic ap-

proaches are widely used in both deep learning models (Goodfellow et al., 2016) and computational models of brain function (Doya et al., 2007). One disadvantage of probabilistic models is that exact inference and learning on distributed representations is often intractable and forces approximate approaches like sampling-based or variational methods (Rezende et al., 2014). In this work, we adopt a modular BCPNN architecture, previously used in abstract models of associative memory (Sandberg et al., 2002; Lansner et al., 2009), action selection (Berthet et al., 2012), and in application to brain imaging (Benjaminsson et al., 2010; Schain et al., 2013) and data mining (Orre et al., 2000). Spiking versions of BCPNN have also been used in biologically detailed models of different forms of cortical associative memory (Lundqvist et al., 2011; Fiebig & Lansner, 2017; Tully et al., 2014). The modules in BCPNN, referred to as hypercolumns (HCs), comprise a set of functional minicolumns (MCs) that compete in a soft-winner-take-all manner. The abstract view of a HC in this abstract cortical-like network is that it represents some attribute, e.g. edge orientation, in a discrete coded manner. A minicolumn comprises a unit that conceptually represents one discrete value (a realization of the given attribute) and, as a biological parallel, it accounts for a local subnetwork of around a hundred recurrently connected neurons with similar receptive field properties (Mountcastle, 1997). Such an architecture was initially generalized from the primary visual cortex, but today has more support also from later experimental work and has been featured in spiking computational models of cortex (Rockland, 2010; Lansner, 2009).

Finally, in this work we highlight additional mechanisms of bias regulation and structural plasticity, introduced recently to the BCPNN framework (Ravichandran et al., 2020), which enable unsupervised learning of hidden representations. The bias regulation mechanism ensures that the activities of all units in the hidden layer are maintained near their target activity by regulating their bias parameter. Structural plasticity learns a set of sparse connections from the input layer to hidden layer by maximizing a local greedy information theoretic score.

## 2 Related Works

A popular unsupervised learning approach is to train a hidden layer to reproduce the input data as, for example, in AE and RBM. The AE and RBM networks trained with a single hidden layer are relevant here since learning weights of the input-to-hidden-layer connections relies on local gradients, and the representations can be stacked on top of each other to extract hierarchical features. However, stacked autoencoders and deep belief nets (stacked RBMs) have typically been used for pre-training procedures followed by end-to-end supervised fine-tuning (using backprop) (Erhan et al., 2009). The recently proposed KH model (Krotov & Hopfield, 2019) addresses the problem of learning solely with local gradients by learning hidden representations only using an unsupervised method. In this network the input-to-hidden connections are trained and additional (non-plastic) lateral inhibition provides competition within the hidden layer. For evaluating the representation, the weights are frozen, and a linear classifier trained with labels is used for the final classification. Our approach shares some common features with the KH model, e.g. learning hidden representations solely by unsupervised methods, and evaluating the representations by a separate classifier (Illing et al. (2019) provides an extensive review of methods with similar goals).

All the aforementioned models employ either competition within the hidden layer (KH), or feedback connections from hidden to input (RBM and AE). The BCPNN uses only the feedforward connections, along with an implicit competition via a local softmax operation, the neural implementation of which would be lateral inhibition.

It is also observed that, for unsupervised learning, having sparse connectivity in the feedforward connections performs better than full connectivity (Illing et al., 2019). In addition to the unsupervised methods, networks employing supervised learning like convolutional neural networks (CNNs) force a fixed spatial filter to obtain this sparse connectivity (Lindsay, 2020). The BCPNN model takes an alternate approach where, along with learning the weights of the feedforward connections, which is regarded as biological synaptic plasticity, a sparse connectivity between the input and hidden layer is learnt simultaneously, in analogy with the structural plasticity in the brain (Butz et al., 2009).

## 3 BAYESIAN CONFIDENCE PROPAGATION NEURAL NETWORK

Here we describe the BCPNN network architecture and update rules (Sandberg et al., 2002; Lansner et al., 2009). The feedforward BCPNN architecture contains two layers, referred to as the input layer and hidden layer. A layer consists of a set of HCs, each of which represents a discrete random variable $X_i$ (upper case). Each HC, in turn, is composed of a set of MCs representing a particular value $x_i$ (lower case) of $X_i$. The probability of $X_i$ is then a multinomial distribution, defined as $p(X_i = x_i)$, such that $\sum_{x_i} p(X_i = x_i) = 1$. In the neural network, the activity of the MC is interpreted as $p(X_i = x_i)$, and the activities of all the MCs inside a HC sum to one.

Since the network is a probabilistic graphical model (see Fig. 1), we can compute the posterior of a target HC in the hidden layer conditioned on all the source HCs in the input layer. We will use $x$'s and $y$'s for referring the HCs in the input and hidden layer respectively. Computing the exact posterior $p(Y_j|X_1, ..., X_N)$ over the target HC is intractable, since it scales exponentially with the number of units. The naive Bayes assumption $p(X_1, .., X_N|Y_j) = \prod_{i=1}^{N} p(X_i|Y_j)$ allows us to write the posterior as follows:

$$p(Y_j|X_1, ..., X_N) = \frac{p(Y_j) \prod_{i=1}^{N} p(X_i|Y_j)}{p(X_1, ..., X_N)} \propto p(Y_j) \prod_{i=1}^{N} p(X_i|Y_j) \tag{1}$$

When the network is driven by input data $\{X_1, .., X_N\} = \{x_1^D, .., x_N^D\}$, we can write the posterior probabilities of a target MC in terms of the source MCs as:

$$p(y_j|x_1^D, ..., x_N^D) \propto p(y_j) \prod_{i=1}^{N} p(x_i^D|y_j) = p(y_j) \prod_{i=1}^{N} \prod_{x_i} p(x_i|y_j)^{\mathbb{I}(x_i = x_i^D)} \tag{2}$$

where $\mathbb{I}(\cdot)$ is the indicator function that equals 1 if its argument is true, and zero otherwise. We have written the posterior of the target MC as a function of all the source MCs (all $x_i$'s). The log posterior can be written as:

$$\log p(y_j|x_1^D, ..., x_N^D) \propto \log p(y_j) + \sum_{i=1}^{N} \sum_{x_i} \mathbb{I}(x_i = x_i^D) \log p(x_i|y_j) \tag{3}$$

Since the posterior is linear in the indicator function of data sample, $\mathbb{I}(x_i = x_i^D)$ can be approximated by its expected value defined as $\pi(x_i) = p(x_i = x_i^D)$. Except for $\pi(x_i)$, all the terms in the posterior are functions of the marginals $p(y_j)$ and $p(x_i, y_j)$. We define the terms bias $\beta(y_j) = \log p(y_j)$ and weight $w(x_i, y_j) = \log p(x_i|y_j)$ in analogy with artificial neural networks.

The inference step to calculate the posterior probabilities of the target MCs conditioned on the input sample is given by the activity update equations:

$$h(y_j) = \beta(y_j) + \sum_{i=1}^{N} \sum_{x_i} \pi(x_i) w(x_i, y_j) \tag{4}$$

$$\pi(y_j) = \frac{\exp(h(y_j))}{\sum_k \exp(h(y_k))} \tag{5}$$

where $h(y_j)$ is the total input received by each target MC from which the posterior probability $\pi(y_j) = p(y_j|x_1^D, ..., x_N^D)$ is recovered by softmax normalization of all MCs within the HC.

As each data sample is presented, the learning step updates the marginal probabilities, weights, and biases as follows:

A

B

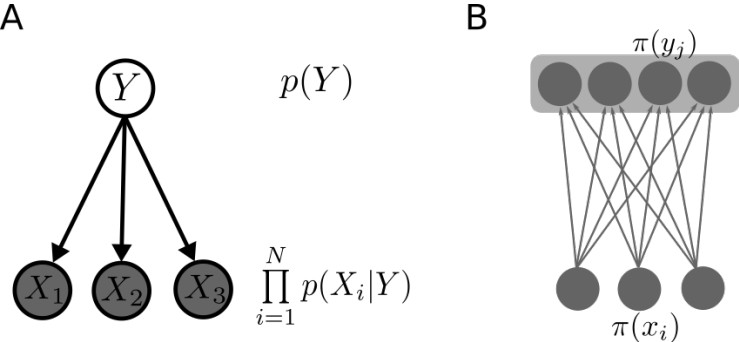

Figure 1: Schematic of the BCPNN architecture with three input HCs and one hidden HC. **A**. The probabilistic graphical model illustrating the generative process with each random variable (circle) representing a HC. The input HCs are observable (shaded circle) and the hidden HC is latent (open circle). The naive Bayes assumption renders the likelihood of generating inputs factorial. **B**. The equivalent neural network where the input HCs are binary (gray circle), hidden HC is multinomial (gray box), and hidden MCs within the HC are the discrete values of the hidden variable (gray circle inside the box).

$$\tau_p \frac{dp(y_j)}{dt} = \pi(y_j) - p(y_j) \tag{6}$$

$$\tau_p \frac{dp(x_i, y_j)}{dt} = \pi(x_i)\pi(y_j) - p(x_i, y_j) \tag{7}$$

$$\beta(y_j) = k_\beta \log p(y_j) \tag{8}$$

$$w(x_i, y_j) = \log \frac{p(x_i, y_j)}{p(y_j)} \tag{9}$$

The terms $\tau_p$ is a learning time constant and $k_\beta$ is the bias gain. The set of Equations 4-9 define the update and learning equations of the BCPNN architecture. In this work, we use the abstract non-spiking model of BCPNN for the purpose of representation learning. The network for unsupervised representation learning requires, in addition to the update and learning equations, the following two mechanisms to enable learning representations (Ravichandran et al., 2020).

### 3.1 BIAS REGULATION

The BCPNN update rule implements Bayesian inference if the parameters are learnt with the source and target layer probabilities available as observations. When the target layer is hidden, we are learning the representations, and we cannot expect the update rule to follow Bayesian inference. In fact, we can see that performing learning and inference simultaneously is counter-productive in this case. Consider a hidden representation with random initialization that assigns some MCs with slightly higher marginal probability $p(y_j)$ than others. Learning would then amplify this difference and find parameters that would associate more input samples with the MCs with high $p(y_j)$, causing the marginals to increase further. One way to circumvent this effect is to promote MCs with low $p(y_j)$ to be more active in the future, like an activity dependent homeostasis process in biological terms (Turrigiano & Nelson, 2004).

We use a bias regulation mechanism, where the bias gain $k_\beta$ for each MC (equal to 1 if only Bayesian inference is performed) depends on $p(y_j)$. One motivation for choosing the bias gain is to influence the marginal $p(y_j)$ alone without affecting the weight parameters that are responsible for learning the input to hidden mapping. The value of $p(y_j)$ is compared with respect to the maximum entropy probability, $p_{MaxEnt} = 1/N_{MC}$, where $N_{MC}$ is the number of MCs per HC. It is worth noting that the maximum entropy is the ideal representation without the input layer since all the MCs have equal marginal probability, and hence acts as the homeostatic reference for bias regulation. The dynamic update of $k_\beta$ with the time constant $\tau_\beta$ follows Eq. 10

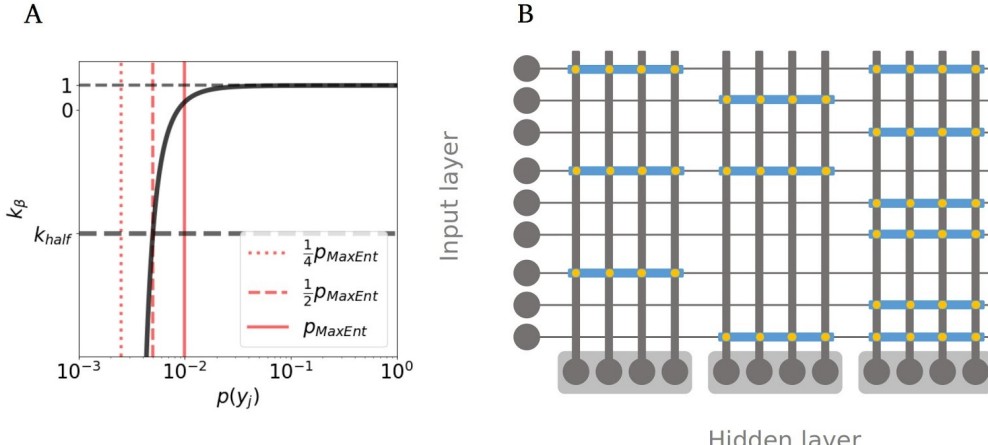

Figure 2: **A**: Bias regulation mechanism. For generating the figure, $k_{half}$=-5 and $p_{MaxEnt}$=0.01 was used. **B**: The schematic of the network used for unsupervised learning. In this network, the input layer contains nine binary HCs (grey circles on the left), and the hidden layer contains three HCs (grey boxes), each of which contains four MCs (grey circles inside the boxes). The existence of a connection between an input HC and hidden HC is shown as a blue strip, i.e., $M_{ij}$=1. The input-hidden weights are shown as yellow dots and are present only when a connection already exists.

$$\tau_\beta \frac{dk_\beta}{dt} = 1 + (k_{half} - 1)\left(\frac{\frac{p_{MaxEnt}}{4}}{p(y_j) - \frac{p_{MaxEnt}}{4}}\right)^2 - k_\beta \tag{10}$$

The mechanism maintains the value of gain $k_\beta$ at around 1 when $p(y_j) \gg \frac{p_{MaxEnt}}{4}$, and drops sharply to negative values when $p(y_j)$ is below $p_{MaxEnt}$ (see Fig. 2A). The rate of this drop is controlled using the hyperparameter $k_{half}$, defined as the value of gain $k_\beta = k_{half}$ at $p(y_j) = \frac{p_{MaxEnt}}{2}$.

## 3.2 STRUCTURAL PLASTICITY

Structural plasticity builds a set of receptive fields for the hidden layer from the input. We define a Boolean variable $M_{ij}$ for the connection from the $i$th input HC to $j$th hidden HC as active, $M_{ij} = 1$, or silent, $M_{ij} = 0$. Each $M_{ij}$ is initialized randomly with probability $p_M$, where setting $p_M$ to a low value ensures patchy and sparse connectivity (Fig. 2B). Once initialized, the total number of active incoming connections to each hidden HC is fixed whereas the outgoing connections from a source HC can be changed. The mutual information (MI) between the $i$th input HC and $j$th hidden HC is estimated from the BCPNN weights: $I_{ij} = \sum_{x_i} \sum_{y_j} P(x_i, y_j) w(x_i, y_j)$. Each input HC normalizes the MI by the total number of active outgoing connections:

$$\hat{I}_{ij} = \frac{I_{ij}}{1 + \sum_k M_{ik}} \tag{11}$$

Since the total number of active incoming connections is fixed, each hidden HC greedily maximizes the $\hat{I}_{ij}$ it receives by removing the active connection with the lowest $\hat{I}_{ij}$ (set $M_{ij}$ from 1 to 0) and adds an inactive connection with the highest $\hat{I}_{ij}$ (set $M_{ij}$ from 0 to 1). We call this operation a flip and use a parameter $N_{flips}$ to set the number of flips made per training epoch.

Table 1: BCPNN model parameters

| Symbol | Value | Descrption |
|--------|-------|------------|
| $N_{HC}$ | 30 | Number of HCs in hidden layer |
| $N_{MC}$ | 100 | Number of MCs per HC in hidden layer |
| $\Delta t$ | 0.01 | Time-step |
| $\mu$ | 10 | Mean of poisson distribution for initializing MCs |
| $N_{sample}$ | 5 | Number of time-steps per sample |
| $N_{epoch}$ | 5 | Number of epochs of unsupervised learning |
| $k_{half}$ | -100 | Bias gain when marginal is $p_{MaxEnt}/2$ |
| $\tau_p^0$ | 0.1 | Multiplier for learning time-constant |
| $\tau_k^0$ | 0.1 | Multiplier for bias gain time-constant |
| $p_M$ | 0.1 | Probability of connections from input to hidden layer |
| $N_{flips}$ | 16 | Number of flips per epoch for structural plasticity |

## 4 EXPERIMENTS

Here we describe the experimental setup for the BCPNN and three other related models for unsupervised learning, as discussed in section 2. Next, we introduce a supervised classification layer trained on the representations learnt by the four models. Finally, we qualitatively study these representations and provide quantitative performance results of the models in supervised classification.

We ran the experiments on the MNIST handwritten digits dataset (LeCun, 1998). MNIST contains $N_{train} = 60000$ training and $N_{test} = 10000$ test images of 28x28 handwritten digits. The images were flattened to 784 dimensions and the grey-scale intensities were normalized to the range [0,1]. The images act as the input layer for the models.

### 4.1 MODELS

We considered four network architectures: BCPNN (c.f. section 3), AE, RBM and, KH. All the models had one hidden layer and 3000 hidden units.

**BCPNN** The BCPNN network had a hidden layer with 30 HCs and 100 MCs per HC. Each sample was clamped to the input layer for $N_{sample}$ iterations of time-step $\Delta t$, and the training was performed for $N_{epoch}$ epochs of the training set. The time constants $\tau_k^0$ and $\tau_p^0$ were scaled by the total training time per epoch, that is, $\tau_k = \tau_k^0 N_{train} N_{sample} \Delta t$ and $\tau_p = \tau_p^0 N_{train} N_{sample} \Delta t$. For tuning the parameters, $\tau_k^0$, $\tau_p^0$, $k_{half}$ and, $N_{flips}$, we used a held-out validation set of 10000 samples from the training set, and chose values that maximize the validation accuracy (for details, see Ravichandran et al. (2020)). The entire list of parameters and their values are listed in Table 1. The simulations were performed on code parallelized using MPI on 2.3 GHz Xeon E5 processors and the training process took approximately two hours per run.

**KH** The KH network was reproduced from the original work using the code provided by (Krotov & Hopfield, 2019). We kept all the parameters as originally described, except for having 3000 hidden units instead of 2000, to be consistent in the comparison with other models.

**RBM** For the RBM network, we used sigmoidal units for both input and hidden layer. The weights were trained using the Contrastive Divergence algorithm with one iteration of Gibbs sampling (CD-1) (Hinton, 2012). The learning rate $\alpha$ was set as 0.01 and the training was done in minibatches of 256 samples for 300 epochs.

**AE** For the AE network, we used sigmoidal units for both hidden layer and reconstruction layer and two sets of weights, one for encoding from input to hidden layer and another for decoding from hidden to reconstruction layer. The weights were trained using the Adam optimizer and L2 reconstruction loss with an additional L1 sparsity loss on the hidden layer. The sparsity loss coeffi-

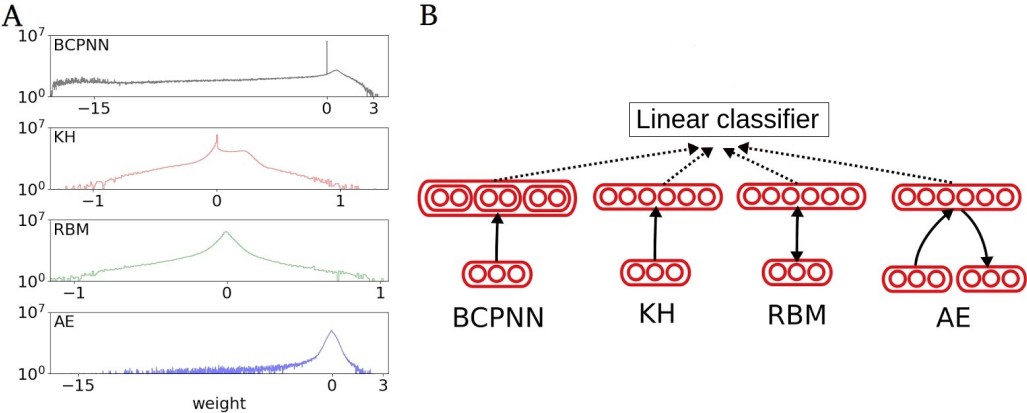

Figure 3: **A**. Histogram of weights from the input layer to hidden layer. The horizontal axis has the minimum to maximum value of the weights as the range, and the vertical axis is in log scale. **B**. Schematic of the four unsupervised learning models under comparison and the supervised classifier. The dotted lines imply we use the representations of the hidden layer as input for the classifier.

cient was determined as $\lambda$=1e-7 by maximizing the accuracy of a held-out validation set of 10000 samples. The training was in minibatches of 256 samples for 300 epochs.

## 4.2 RECEPTIVE FIELD COMPARISON

As can be observed in Fig. 3A, the distribution of weight values considerably differs across the networks examined in this work. It appears that the range of values for BCPNN corresponds to that reported for AE, whereas for KH and RBM, weights lie in a far narrower interval centered around 0. Importantly, BCPNN has by far the highest proportion of zero weights (90%), which renders the connectivity truly sparse.

In Fig. 4, we visualize the receptive fields of the four unsupervised learning networks. Firstly, it is straightforward to see that the receptive fields of all the networks differ significantly. The RBM (Fig. 4C) and AE (Fig. 4D) have receptive fields that are highly localized and span the input space, a characteristic of distributed representations. The KH model (Fig. 4B) has receptive fields that resemble the entire image, showing both positive and negative values over the image, as a result of Hebbian and anti-Hebbian learning Krotov & Hopfield (2019). Generally, local representations like mixture models and competitive learning, as opposed to distributed representations, tend to have receptive fields that resemble prototypical samples (Rumelhart & Zipser, 1985). With this distinction in mind, the receptive fields in the BCPNN should be closely examined (Fig. 4A). The receptive fields of HCs (first column) are localized and span the input space, much like a distributed representation. Within each HC however, the MCs have receptive fields (each row) resembling prototypical samples, like diverse sets of lines and strokes. This suggests that the BCPNN representations are "hybrid", with the higher-level HCs coding distributed representation, and the lower level MCs coding local representation.

## 4.3 CLASSIFICATION PERFORMANCE

For all the four models of unsupervised learning, we employed the same linear classifier for predicting the labels (see Fig. 3B). This allowed us to consistently evaluate the representations learned by the different models. The linear classifier considers the hidden layer as the input and the MNIST labels as the output. The output layer consists of softmax units for the 10 labels. The classifier's weights were trained by stochastic gradient descent with the Adam optimizer (Kingma & Ba, 2014) using cross-entropy loss function. The training procedure used minibatches of 256 samples and a total of 300 training epochs.

The results of the classification are shown in Table II. All the results presented here are the mean and standard deviation of the classification accuracy over ten random runs of the network. We performed

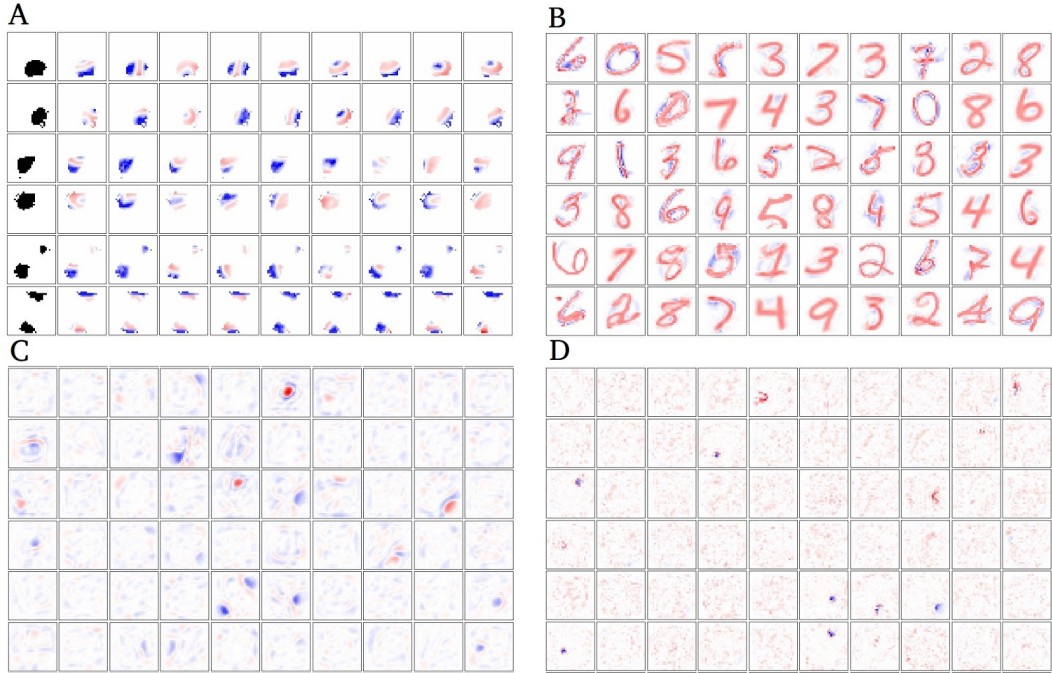

Figure 4: Receptive fields of different unsupervised learning methods. For each model, the positive and negative values are normalized, such that blue, white, and red represent the lowest, zero, and highest value of weights. **A. BCPNN**: Each row corresponds to a randomly chosen HC and the constituent MCs of BCPNN. First column shows the receptive field of HC (black means $M_{ij}$=1). The remaining columns show the receptive field of nine randomly chosen MCs out of 100 MCs within the HC. **B. KH**, **C. RBM**, **D. AE**: Receptive fields of 60 randomly chosen hidden units out of 3000.

Table 2: Accuracy comparison

| Model | Tuned parameters | Accuracy (train) | Accuracy (test) |
|-------|------------------|------------------|-----------------|
| BCPNN | $\tau_p^0$=0.1, $\tau_k^0$=0.1, $k_{half}$=-100 | $100.00 \pm 0.00$ | $97.77 \pm 0.12$ |
| KH | See Krotov & Hopfield (2019) | $98.75 \pm 0.01$ | $97.39 \pm 0.06$ |
| RBM | $\alpha = 0.01$ | $98.92 \pm 0.04$ | $97.67 \pm 0.10$ |
| AE | $\lambda$ =1e-7 | $100.00 \pm 0.00$ | $97.78 \pm 0.09$ |

three independent comparisons of BCPNN with KH, RBM, and AE using the Kruskal-Wallis H test. BCPNN outperforms KH[1] ($p$=0.02), while there is no statistical difference with RBM ($p$=0.28) / AE ($p$=0.30).

## 5 DISCUSSION

We have evaluated four different network models that can perform unsupervised representation learning using correlation based biologically plausible local learning rules. We made our assessment relying on the assumption that the saliency of representations is reflected in their class dependent separability, which can be quantified by classification performance, similar to Illing et al. (2019) and Krotov & Hopfield (2019). Learning representations without supervised fine-tuning is a harder task

---

[1]This is lower than the test accuracy of 98.54% reported by Krotov & Hopfield (2019). This is due to differences in the classifier used: exponentiated ReLU activation along with an exponentiated loss function. We instead used a simpler softmax activation and cross-entropy loss.

compared to similar networks with end-to-end backprop, since the information about the samples' corresponding labels cannot be utilized. Consequently, representations learnt with unsupervised methods cannot be expected to offer better class separability than the classification performance reported by supervised end-to-end approaches. We show that the investigated unsupervised methods score remarkably similar around 97.7%, which is only slightly worse compared to the 98.5% accuracy of networks with one hidden layer trained with end-to-end backprop (LeCun et al., 1998).

We also showed that the recently proposed BCPNN model performs competitively against other unsupervised learning models. The modular structure of the BCPNN layer led to "hybrid" representations that differ from the well-known distributed and local representations. In contrast to the minibatch method of other unsupervised learning methods, learning in BCPNN was chosen to remain incremental using dynamical equations, since such learning is biologically feasible and useful in many autonomous engineering solutions. Despite the slow convergence properties of an incremental approach, BCPNN required only 5 epochs of unsupervised training, in comparison to 300 epochs for AE and RBM, and 1000 epochs for KH. The incremental learning, along with modular architecture, sparse connectivity, and scalability of BCPNN is currently also taken advantage of in dedicated VLSI design (Stathis et al., 2020).

One important difference between current deep learning architectures and the brain concerns the abundance of recurrent connections in the latter. Deep learning architectures rely predominantly on feedforward connectivity. A typical cortical area receives only around 10% of synapses from lower order structures, e.g. thalamus, and the rest from other cortical areas (Douglas & Martin, 2007). These feedback and recurrent cortical connections are likely involved in associative memory, constraint-satisfaction e.g. for figure-ground segmentation, top-down modulation and selective attention (Douglas & Martin, 2007). Incorporating these important aspects of cortical computation can play a key role in improving our machine learning models and approaches.

It is important to note that the unsupervised learning models discussed in this work are proof-of-concept designs and not meant to directly model some specific biological system or structure. Yet, they may shed some light on the hierarchical functional organization of e.g. sensory processing streams in the brain. Further work will focus on extending our study to multi-layer architectures.

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
