# OpenReview forum: "Brain-like approaches to unsupervised learning of hidden representations - a comparative study "
_ICLR.cc/2021/Conference — Reject_

### Official Review · AnonReviewer3 · 2020-10-27
**This is a comparative study of four unsupervised learning approaches. The authors specifically focused on the brain-like BCPNN model. Overall, the methods were clear and fair. However, it is difficult to draw reliable conclusions based on current comparison results.**

**Rating:** 6
**Confidence:** 3

**Review:**

This paper evaluated four unsupervised learning approaches (BCPNN, KH, RBM, AE) by training a supervised classification layer on top of the hidden representation. Specifically, the authors qualitatively compared the receptive fields and quantitatively compared the classification performance across four models. The authors emphasized the advantages of BCPNN since it applies biologically plausible local learning rules and requires fewer epochs for convergence.

Overall, the comparison was fair and solid. The description of the BCPNN model in section 3 was clear and comprehensive. But the authors did not provide sufficient details of key mechanisms in the other three models (especially the KH model, which also used brain-like learning rules). The detailed introduction of the other three models should be an important component since this is a comparative study. The results were clearly stated, but the insignificant difference in the classification accuracy comparison (Table 2) can hardly lead to a reliable conclusion about which unsupervised method is better. And it would be better if the authors could provide more interpretations about the "hybrid" receptive fields of HCs and MCs in BCPNN (Fig. 3A).

##### Specific comments and questions:
1. In the original KH model (Krovtov & Hopfield, 2019), they also tested the classification accuracy on the MNIST dataset, and the result reached an error rate of 1.46% with 2000 hidden states. This is better than the reproduced result shown here (97.39% accuracy) with 3000 hidden states. Is this accuracy drop caused by a different setting of hyperparameters? Then is it fair to say that "BCPNN outperforms KH"?
2. Could you provide more explanation on Eq. 11? Why this dynamic update of $k_{\beta}$ could be used as a desired bias regulation?
3. When the number of total hidden units is fixed, what would be the effect of changing the ratio $\frac{N_{MC}}{N_{HC}}$ in BCPNN?
4. The "hybrid" structure of BCPNN provides interesting receptive field results in Fig. 3A. Is this structure generalizable to a model with multiple hidden layers?

##### Minor:
Page 4. in 3.1 Bias regulation. Typo: Eq. 6 should be Eq. 11. And "the value of gain $k_{\beta}$ at around 1 when $P(y_j) - \frac{p_{MaxEnt}}{4}$," missing $\gg 0$ here?

---

> ### Author Response · Authors · 2020-11-19
> **Response to AnonReviewer3**
>
> We thank the reviewer for the comments and valuable feedback.
>
> The KH model as described in the original work is quite detailed and a proper description may not fit within this article. We therefore stated the key principles involved in the “Related Works” section. As for AE and RBM, we assumed prior knowledge of these models, since they are widely used and by now a standard in machine learning. It is true that our comparisons of BCPNN with KH, RBM, and AE showed BCPNN outperforms KH, while there is no difference with RBM and AE. However, our objective is to compare not just the accuracy performance, but also the nature of the representations learnt (for instance receptive fields).
>
> 1: Thank you for pointing it out! The original KH model indeed reported higher accuracy compared to our version of the same model. This is because in the KH model, for the supervised classifier on the learned representation, a non-linear model (exponentiated ReLU activation along with an exponentiated loss function) was used. We decided to use the same simple classifier (with softmax activation and cross-entropy loss) for all the models evaluated as we were interested in hidden representations being label-wise linearly separable. We expected this to give a fair comparison. A short version of this comment is added as a footnote in the manuscript.
>
> 2: This bias regulation is related to the so-called “intrinsic plasticity” of cortical neurons. It is known that they can adapt their baseline activation based on previous activity [see e.g. Fransén, E., Tahvildari, B., Egorov, A. V., Hasselmo, M. E., and Alonso, A. A. (2006). Mechanism of graded persistent cellular activity of entorhinal cortex layer V neurons. Neuron 49, 735–746. doi: 10.1016/j.neuron.2006.01.036]. In a more detailed spiking model in the NEST simulator, it has been implemented as an adaptive potassium channel. Different neuron types can be expected to do this in different ways. Though the kind of regulation we describe here is hypothetical, it is local computation within the neuron and can be considered biologically plausible.
>
> 3: We will investigate the effects of changing this ratio as well as size of the hidden layer. However, we won’t be able to include it in this work.
>
> 4: The “hybrid” representations are directly the result of the BPCNN modular architecture. Since different hypercolumns focus on different parts of the image and learn features over the receptive field, they form a distributed representation. The minicolumns within each hypercolumn learn prototypical samples (clustering). We consider this modular architecture of the layer to be generalizable to multiple layers in the cortical hierarchy. The columnar circuit (canonical microcircuit) and architecture (minicolumns and hypercolumns) is considered to be conserved throughout the cortex since the work of Mountcastle [Mountcastle, V.B., 1997. The columnar organization of the neocortex. Brain: a journal of neurology, 120(4), pp.701-722].

---

### Official Review · AnonReviewer2 · 2020-10-28
**The paper systematically investigates Bayesian Confidence Propagating Neural Networks (BCPNN) on learning unsupervised representations on MNIST dataset. It presents a comprehensive comparison of four different commonly used unsupervised methods.**

**Rating:** 7
**Confidence:** 3

**Review:**

Systematic investigation of biologically inspired algorithms and architectures is a very important research topic.

The paper investigates Bayesian Confidence Propagating Neural Networks (BCPNN) on learning unsupervised representations on MNIST dataset. It presents a comprehensive comparison of four different commonly used unsupervised methods.

The strong merit of BCPNN approach is the nice receptive fields of the hypercolumns (HC) and minicolumns (MC) learned by the proposed algorithm. As the authors point out, the advantage of the proposed algorithm is that it is able to produce sparse and highly localized (in the image plane) receptive fields for the MCs. Also, those filters look much cleaner than the counterparts of the classical algorithms considered in figure 3.  Additionally, the authors demonstrate that their representations stand in line with previously published proposals in terms of the classification accuracy.

The main weakness of this work is that the proposed method has only been investigated on MNIST and in one-layer architectures. At the same time, given the novelty of the approach, I think it deserves attention even in this simplest setting considered in this manuscript.

Small comments:
1. Section4 (3rd line) should be “learned”
2. There are some misprints around equation 11, such as the use of p(y) vs. P(y) is inconsistent. Also, it seem that the word “large” is missing following the formula in the line after equation 11.
3. I also find panel B in figure 2 confusing. The way it is presented makes it look like the authors have combined the outputs of four networks to feed into the classifier, while in reality those four networks were evaluated one by one.
4. It would be better to designate a new variable for the left hand side of equation 12, since I_ij is already taken.

Post Rebuttal:

Thank you for the response. I have read the discussion with other reviewers. A small comment: while I agree that it is reasonable to keep the classifier the same for all the models (softmax with cross entropy) for a fair comparison, I disagree that the activation function for the first layer should be kept as ReLU in the KH model. In fact KH explain that this is a suboptimal choice in Fig. 4 of their original paper. Using powers of ReLUs should increase the KH accuracy. Overall, I think that this is a nice paper, and I am inclined to keep my initial score.

---

> ### Author Response · Authors · 2020-11-19
> **Response to AnonReviewer2**
>
> We thank the reviewer for the comments and valuable feedback.
>
> We acknowledge that one-layer architecture with MNIST is quite limited. However, given the nature of constraints handled: unsupervised learning with local learning rules, there is little prior work that accomplishes the same performance. We are also working on applying our model to harder datasets and extending the network to deeper architecture.
>
> Point 1,2,3,4: Corrected

---

### Official Review · AnonReviewer4 · 2020-11-03
**An incremental contribution lacking convincing experiments**

**Rating:** 4
**Confidence:** 4

**Review:**

Summary:
The Bayesian Confidence Propagating Neural Network has recently been extended to the case of unsupervised learning (Ravichandran et al., IJCNN, 2020). This paper compares this extension to restricted Boltzmann machines, autoencoders, and a biologically plausible model proposed by (Krotov & Hopfield, PNAS, 2019) on the MNIST dataset. For evaluation the authors consider the learned receptive fields and the classification performance of a linear classifier. The paper is very similar to (Ravichandran et al., IJCNN, 2020) but with an extended experimental section.


Positives:

+ Biologically plausible methods for unsupervised learning are an interesting area of research.

+ There has been relatively little research on structural plasticity.


Concerns:

- The paper does not introduce anything new but merely compares existing methods.

- The comparison is not an extensive study, but limited to one dataset, MNIST, and few alternative proposals, of which only the KH model is deemed 'brain-like'.

- There seems to be something off with the experimental results. Krotov & Hopfield report a better test accuracy of 98.54% in their original paper in spite of using less hidden units.

- BCPNN's performance is mediocre. It is even outperformed by random shallow networks with fixed, localized, random & random Gabor filters in the hidden layer (Illing et al, Neural Networks, 2019)

- Lacking performance could be excused by greater biological plausibility as a neuroscientific contribution, which is however not the case here. As the authors themselves state, their model is 'abstract' (page 4) and not a neural implementation but merely 'uses implicit competition via a local softmax operation, the neural implementation of which would be lateral inhibition'.


Minor comments:

$\pi(x_i)$ in Eq (6) is never introduced.

The line above Eq (11) should probably refer to (11) not (6).

There's a typo in the sentence after Eq (11).

The hybrid representation might be interesting. Is it any more biological than the well-known distributed and local representations?

---

> ### Author Response · Authors · 2020-11-19
> **Response to AnonReviewer4**
>
> We thank the reviewer for the review and valuable feedback.
>
> 1,2: This works concerns evaluating different models with the criteria of “brain-like” as we define it in terms of local learning rules and unsupervised learning. We do acknowledge that recent work, some of which we cited, do a more extensive study of such methods. However, our objective here is to compare not just the accuracy performance, but also the nature of the representations learnt (for eg. receptive fields), specifically for unsupervised learning methods. In our comparative study, we also included popular neural network models like RBMs and AEs as they are unsupervised models with local learning. We acknowledge the limitation of evaluating only on MNIST, but extending such brain-like neural networks to harder datasets is still an active area of research and we work on it.
>
> 3: Thank you for pointing it out! The original KH model indeed reported higher accuracy compared to our version of the same model. This is because in the KH model, for the supervised classifier on the learned representation, a non-linear model (exponentiated ReLU activation along with an exponentiated loss function) was used. We decided to use the same simple classifier (with softmax activation and cross-entropy loss) for all the models evaluated as we were interested in hidden representations being label-wise linearly separable. We expected this to give a fair comparison. A short version of this comment is added as a footnote in the manuscript.
>
> 4: It is true that Illing et al. (2019) showed random projections and local random Gabor filters outperform many single-layer unsupervised as well as supervised learning models like backprop and feedback-alignment. As we mentioned earlier, our objective here is to compare not just the accuracy performance, but also the nature of the representations learnt. Also, local projections would be limited to image-like data with a local correlation structure, since localized receptive fields can only be hard-coded for such data. All the models we compare do not make this restriction for learning the weight parameters (and structural plasticity) and are thus more general in nature. Our model with structural plasticity would find the pixel dependencies and give the same performance even if MNIST images were transformed by a permutation operation - the same for all images. In for instance olfactory sensory data there is not a topology like in e.g. the visual system. A biologically plausible learning architecture should be able to handle such data as well.
>
> 5: The normalization within a hypercolumn using a softmax operation is not a drawback or biologically unrealistic as we see it. Mapping our model to the cortex, activity of each minicolumn represents the probability of an event and is the average firing rate of around hundred excitatory neurons (pyramidal neurons), while the inhibitory neurons  (basket cells) project locally to all the minicolumns within the same hypercolumn and provide local lateral inhibition and competition. In the visual system it has been suggested to provide local divisive normalization [see e.g. Carandini M, Heeger DJ, Movshon JA. Linearity and normalization in simple cells of the macaque primary visual cortex. J Neurosci. 1997;17(21):8621-8644. doi:10.1523/JNEUROSCI.17-21-08621.1997]. We consider this computation biologically plausible, and chose to model it using a simple softmax operation in the interest of abstraction.
>
> The “hybrid” representations are directly the result of the BPCNN modular architecture. Since different hypercolumns focus on different parts of the image and learn features over the receptive field, they form a distributed representation. The minicolumns within each hypercolumn learn prototypical samples (clustering). We consider this modular architecture of the layer to be generalizable to multiple layers in the cortical hierarchy. The columnar circuit (canonical microcircuit) and architecture (minicolumns and hypercolumns) are considered to be conserved throughout the cortex since the work of Mountcastle [Mountcastle, V.B., 1997. The columnar organization of the neocortex. Brain: a journal of neurology, 120(4), pp.701-722].

---

### Official Review · AnonReviewer5 · 2020-11-06
**Lacking in clarity**

**Rating:** 5
**Confidence:** 4

**Review:**

Summary:
This paper proposes a set of biologically plausible update rules that can be used to compute latent representations. The paper presents a number of heuristics to set hyper-parameters and train the proposed model. The model is compared to RBMs, auto-encoders and another recently proposed biologically-plausible model.

Pros:
- The method explores a new kind of model motivated by having biologically plausible update rules.
- The experimental results include an interesting comparison of the learned features.

Cons:
- The paper places emphasis on interpreting the update rules as Bayesian confidence propagation. Yet, the underlying probabilistic graphical model is not clearly described. It would be useful to have a figure in the paper that describes the model, including whether the model is directed or undirected, which variables are observed or unobserved, if the model is directed what is the generative model, etc.
- Two assumptions are mentioned about the graphical model : \\( P(X_1,..,X_N) = \prod_i P(X_i) \\), and \\( P(X_1,..,X_N|Y_j) = \prod_i P(X_i|Y_j) \\). Unless I misunderstood, the first assumption is saying that each dimension in the input data \\(X\\) is independent of the others. This is a very strong assumption and makes the model weak. The paper does not include an explanation about why these assumptions make sense, or how they influence the model's inductive bias relative to other probabilistic models.
- The derivation of the update rules is also unclear. The approximation that the indicator function \\(I(x_i=x^D_i)\\) can be replaced by its expected value \\(P(x^D_i)\\) is hard to understand. In particular, I could not understand how to parse Eq (4) which has a sum over \\(x_i\\). Is this still dependent on the input, given that the indicator function has been replaced by its expectation ? It would be good to have a more clear derivation of the update rules starting from the probabilisitic model.
- The paper makes some very general assertions that do not add to the point being made. For example, "One disadvantage of probabilistic models is that the known methods do not scale well in practice." Models such as VAEs are probabilistic models which scale quite well to high-dimensional inputs and large amounts of data.
- In the introduced model, each HC seems to have a similar representational capacity as a softmax hidden unit. Therefore, instead of comparing to RBMs and AEs with sigmoid units, comparisons to RBMs and AEs with softmax hidden units will be more relevant.

Overall, the paper can be improved along two directions : making the probabilistic interpretation more clear (specifying the graphical model clearly, deriving update rules), and doing experiments with softmax hidden units (so that the only thing changing is the update rules, and not the model architecture).

---------------
Post-rebuttal

- Figure 1 is helpful in understanding the model. Thank you for adding that.
- The additional experiments are also appreciated. This seems to indicate that it's not just the architecture but the learning rules that make the BCPNN model work well.
- It would also be helpful to visualize the features learned by softmax units in RBMs and AEs and see if this results in a similar pattern of HCs encoding broad regions and MCs encoding variations within those regions.
- It seems that the RBMs and AEs were not trained using any sparsity penalty. For example, Page 11 of https://www.cs.toronto.edu/~hinton/absps/guideTR.pdf and https://web.stanford.edu/class/cs294a/sparseAutoencoder.pdf. Having a target sparsity can have a significant impact on the learned features. Higher sparsity makes the features look more like stroke like and localized, and less spread out all over the visual field (as they do in Fig 4, C and D).
- Based on the additional experiments, I will be increasing my score to 5. However, given that the main contribution of the paper is a comparative study, the paper can add value by doing a more thorough comparison against variants of AEs and RBMs that have otherwise similar properties (such as keeping the HC-MC (softmax) architecture and sparsity levels the same).

---

> ### Author Response · Authors · 2020-11-19
> **Response to AnonReviewer5**
>
> We thank the reviewer for the comments and valuable feedback.
>
> 1: We now included a figure (Fig. 1) explaining BCPNN in terms of the graphical model.
>
> 2: The derivation of BCPNN does include strong assumptions of factorial likelihood while calculating the posterior probability of hidden variables. This naive Bayes assumption is also comparable to the widely used artificial neuron models (McCulloch-Pitts) that assume inputs to be linearly separable. Although it is certainly the case that input data is neither conditionally independent or linearly separable, these assumptions allow for avoiding intractable computations. Furthermore, the naive Bayes assumption becomes lesser of a problem as we get better estimates of hidden variables.
>
> We removed the second assumption that the input variables are factorial (from the older version of manuscript), as this is unnecessary in computing the posterior (it is absorbed while normalizing with softmax). It was written to be consistent with the previous BCPNN models, but we removed it now to avoid any further confusion.
>
> 3: This assumption is common in probabilistic modeling with neural networks. The indicator function is a binary valued (either 0 or 1), and is replaced with the real-valued data, which is interpreted as p(x). For identical treatment in Boltzmann machines see http://www.scholarpedia.org/article/Boltzmann_machine#Mean_field_Boltzmann_machines
>
> 4: We were not clear about this. What we intended to say was “One disadvantage of a probabilistic model is that doing simple exact inference and learning on distributed representations is intractable and forces approximate solutions (like numerous sampling or variational methods)”. This is now corrected.
>
> 5: We used binary units for RBMs and AEs since it is widely used and well tested. Also, softmax units cannot be used in the KH model since it uses exponentiated ReLU units.

---

> > ### Author Response · Authors · 2020-11-24
> > **Additional experiments regarding point 5**
> >
> > In the experiments, we compare the BCPNN model with modular hidden architecture (hypercolumns and minicolumns) with binary RBMs, binary AEs, and KH model with ReLU units. Following the review, we have also experimented with RBM and AE networks with softmax units to have similar hidden layer representation as BCPNN. The experiment involved fixing the size of the hidden layer to 3000 units, while changing the ratio of number of MCs per HC to the number of HCs. The two tables below show the accuracy (over 10 trials) of the linear classifier when trained on the hidden representations. The results in the first column are from hidden layers with only binary units and we denote this as 3000 : 1. We see that performance of both RBM and AE decreases with the increasing ratio of modularisation (more MCs per HC and fewer HCs). The highest accuracy is obtained for the least modular layer. When compared at the same ratio of 3.34 used for BCPNN (training accuracy of 100±0 and test accuracy of 97.77±0.12) both RBM and AE perform poorly.
> >
> > Table 1: RBM
> >
> > | #HC : #MCs per HC |   3000 : 1   |    1000 : 3  |    500 : 6   |   300 : 10   |   100 : 30   |    50 : 60   |   30 : 100   |
> > |:-----------------:|:------------:|:------------:|:------------:|:------------:|:------------:|:------------:|:------------:|
> > |       Ratio       |    0.00034   |     0.003    |     0.012    |     0.034    |     0.30     |      1.2     |     3.34     |
> > |     train (%)     | 98.92 ± 0.04 | 99.17 ± 0.05 | 98.69 ± 0.07 | 97.28 ± 0.05 | 91.88 ± 0.11 | 83.93 ± 0.34 | 77.59 ± 0.56 |
> > |      test (%)     | 97.67 ± 0.10 | 97.68 ± 0.03 | 97.25 ± 0.05 | 96.14 ± 0.03 | 91.63 ± 0.14 | 84.18 ± 0.35 | 77.94 ± 0.60 |
> >
> > Table 2: AE
> >
> > | #HC : #MCs per HC |    3000 : 1   |    1000 : 3  |    500 : 6   |    300 : 10   |    100 : 30   |     50 : 60    |   30 : 100   |
> > |:-----------------:|:-------------:|:------------:|:------------:|:-------------:|:-------------:|:--------------:|:------------:|
> > |       Ratio       |    0.00034    |     0.003    |     0.012    |     0.034     |      0.30     |       1.2      |     3.34     |
> > |     train (%)     | 100.00 ± 0.00 | 99.56 ± 0.02 | 98.72 ± 0.02 |  98.96 ± 0.18 |  97.74 ± 0.08 | 95.13 ± 0.20   |  93.18 ±0.13 |
> > |      test (%)     |  97.78 ± 0.09 | 97.71 ± 0.09 | 97.68 ± 0.06 | 97.10 ±  0.22 |  96.57 ± 0.20 |  94.44 ± 0.30  | 92.95 ± 0.27 |

---

### Decision · Program_Chairs · 2021-01-07
**Final Decision**

**Decision:**

Reject

**Comment:**

This paper conducts a comparison between a small set of models (4 in total) for unsupervised learning. Specifically, the authors focus on comparing Bayesian Confidence Propagating Neural Networks (BCPNN), Restricted Boltzmann Machines (RBM), a recent model by Krotov & Hopfield (2019) (KH), and auto-encoders (AE). The authors compare trained weight distributions, receptive field structures, and linear classification on MNIST using the learned representations. The first two comparisons are essentially qualitative comparisons, while on classification accuracy, the authors report similar accuracy levels across the models.

This paper received mixed reviews. Reviewers 4 and 5 felt it did not contribute enough for acceptance, while Reviewers 2 & 3 were more positive. However, as noted by a few of the reviewers, this paper does not appear to achieve much, and provides very limited analysis and experiments on the models. It isn't introducing any new models, nor does it make any clear distinctions between the models examined that would help the field to decide which directions to pursue.  The experiments add little insight into the differences between the models that could be used to inform new work. Thus, the contribution provided here is very limited.

Moreover, the motivations in this paper are confused. In general, it is important for researchers at the intersection of neuroscience and machine learning to decide what their goal is when building and or comparing models. Specifically, is the goal: (1) finding a model that may potentially explain how the brain works, or (2) finding better machine learning tools?

If the goal is (1), the performance on benchmarks is less important. However, clear links to experimental data, such that experimental predictions may be possible, are very important. That's not to say that a model must be perfectly biologically realistic to be worthwhile, but it must have sufficient grounding in biology to be informative for neuroscience. However, in this manuscript, as was noted by Reviewer 4, the links to biology are tenuous. The principal claim for biological relevance for all the models considered seems to be that the update rules are local. But, this is a loose connection at best. There are many more models of unsupervised learning with far more physiological relevance that are not considered here (see e.g. Olshausen & Field, 1996, Nature; Zylberberg et al. 2011, PLoS Computational Biology; George et al., 2020, bioRxiv: https://doi.org/10.1101/2020.09.09.290601). It is true that some of these models use non-local information, but given the emerging evidence that locality is not actually even a strict property in real synaptic plasticity (see e.g. Gerstner et al., 2018, Frontiers in Neural Circuits; Williams & Holtmaat, 2018, Neuron; Banerjee et al., 2020, Nature), an obsession with rules that only use pre- and post-synaptic activity is not even clearly a desiderata for neuroscience.

If the goal is (2), then performance on benchmarks, and some comparison to the SotA, is absolutely critical. Yet, this paper does none of this. Indeed, the performance achieved with the four models considered here is, as noted by Reviewer 4, very poor. In contrast, there have been numerous advances in unsupervised (or "self-supervised") learning in ML in recent years (e.g. Contrastive Predictive Coding, SimCLR, Bootstrap Your Own Latent, etc.), all of which achieve far better results than the four models considered here. Thus, the models being compared here cannot inform machine learning, as they do not appear to provide any technical advances. Of course, some models may combine goals (1) & (2), e.g. seeking increased physiological relevance while also achieving decent benchmark performance (see e.g. Sacramento et al., 2018, NeurIPS), but that is not really the situation faced here, as the models considered have little biological plausibility (as noted above) and achieve poor performance at the same time.

Altogether, given these considerations, although this paper received mixed reviews, it is clearly not appropriate for acceptance at ICLR in the Area Chair's opinion.